# Complexes of Cu–Polysaccharide of a Marine Red Microalga Produce Spikes with Antimicrobial Activity

**DOI:** 10.3390/md20120787

**Published:** 2022-12-19

**Authors:** Nofar Yehuda, Levi A. Gheber, Ariel Kushmaro, Shoshana (Mails) Arad

**Affiliations:** 1Avram and Stella Goldstein-Goren Department of Biotechnology Engineering, Ben-Gurion University of the Negev, Beer-Sheva 8410501, Israel; 2The Ilse Katz Institute for Nanoscale Science and Technology, Ben-Gurion University of the Negev, Beer-Sheva 8410501, Israel; 3School of Sustainability and Climate Change, Ben-Gurion University of the Negev, Beer-Sheva 8410501, Israel

**Keywords:** sulfated polysaccharides, red microalgae, metal complexes, antimicrobial activities, spike formation, biomaterials

## Abstract

Metal–polysaccharides have recently raised significant interest due to their multifunctional bioactivities. The antimicrobial activity of a complex of Cu_2_O with the sulfated polysaccharide (PS) of the marine red microalga *Porphyridium* sp. was previously attributed to spikes formed on the complex surface (roughness). This hypothesis was further examined here using other Cu–PS complexes (i.e., monovalent-Cu_2_O, CuCl and divalent-CuO, CuCl_2_). The nanostructure parameters of the monovalent complexes, namely, longer spikes (1000 nm) and greater density (2000–5000 spikes/µm^2^) were found to be related to the superior inhibition of microbial growth and viability and biofilm formation. When *Escherichia coli* TV1061, used as a bioluminescent test organism, was exposed to the monovalent Cu–PS complexes, enhanced bioluminescence accumulation was observed, probably due to membrane perforation by the spikes on the surface of the complexes and consequent cytoplasmic leakage. In addition, differences were found in the surface chemistry of the monovalent and divalent Cu–PS complexes, with the monovalent Cu–PS complexes exhibiting greater stability (ζ-potential, FTIR spectra, and leaching out), which could be related to spike formation. This study thus supports our hypothesis that the spikes protruding from the monovalent Cu–PS surfaces, as characterized by their aspect ratio, are responsible for the antimicrobial and antibiofilm activities of the complexes.

## 1. Introduction

Polysaccharides are well-known due to their wide range of biological activities in various aspects (e.g. medicine, cosmetics, and foodomics) [1]. However, it is well-understood that their bioactivity is dependent on their structure [2,3], which can be enhanced by certain modifications [4]. We have thus assumed that the addition of metals to the sulfated polysaccharide of red microalgae with anion exchange capabilities will enable synergism between the metal and the polysaccharide and will generate complexes with new functional activities.

The Cu_2_O–PS complexes that constitute the subject of this study bring together the antimicrobial properties of Cu_2_O and the nanostructure of the sulfated cell-wall polysaccharide of *Porphyridium* sp. [5,6,7]. This sulfated polysaccharide is harvested from the algal culture medium. It has a molecular mass of 3−7 × 10^6^ Da and is composed of 10 different sugars, the main ones being xylose, glucose, and galactose [6,8]. The polysaccharide also contains glucuronic acid and half-ester sulfate groups (about 9%*w/v* sulfate), which endow it with a negative charge [6,7,8,9]. It is believed that the functions of this sulfated cell-wall polysaccharide are to maintain cell humidity and serve as a free radical scavenger under conditions of high light [6,7,10,11]. Importantly, it is also thought that the sulfated polysaccharide provides a buffer layer around the cells, protecting them not only against severe environmental conditions but also against bacteria, viruses, and fungi [5,6,12,13,14,15]. These unique qualities make it suitable for a variety of applications. In keeping with these ideas, the sulfated polysaccharide of *Porphyridium* sp. has been shown to exhibit a variety of bioactivities including antiviral [14,15], anti-inflammatory [12], antioxidant [11], and antibiofilm [16,17] activities, along with superior bio-lubricant properties of industrial applications including those in the cosmetics, pharmaceutical, and food industries (as a gelling, thickening, and stabilizing agent) [12,18,19,20].

The effective inhibition of microbial growth and biofilm formation by a Cu_2_O–polysaccharide (PS) complex, in which the polysaccharide is produced by the marine red microalga *Porphyridium* sp., was previously reported by us [16]. The complex was particularly efficacious in inhibiting the growth of the fungus *Candida albicans* [16]. In that study, it was suggested that the antimicrobial and antibiofilm activities of the complex could be attributed to the needle-like topographical protrusions—spikes—that formed on the surface of the Cu_2_O−PS complex. This notion was in keeping with previous studies on biocidal materials that showed that their nanostructured surfaces—whether nanopillars, nanopatterns, or other nanostructures—influenced their antimicrobial activities [20,21,22,23]. For example, the microbial adhesion force to the spikes (having an aspect ratio, i.e., the ratio of spike height to the width, of 0.5–4.6 nm) of the nanostructured surface of cicada wings was shown to directly influence the viability of *Saccharomyces cerevisiae* [24]. Similarly, the strong adhesion between the spikes (nanopillars) on the natural nanotopography of dragonfly wings (roughness of 32–41 nm) caused damage to the cell membrane and hence a bactericidal effect on *Escherichia coli* [25]. Indeed, the degree of rigidity of bacterial cells has been found to influence their susceptibility to mechanical damage by nanostructured anti-bacterials, with Gram-negative bacteria being more sensitive to spike-mediated rupture than Gram-positive bacteria [26,27,28]. In parallel, a number of studies have shown that the size and density of nanotopographic spikes influence the efficacy of bactericidal activity [24,27,29,30]. Thus, there is a growing understanding that the surface topography of some antimicrobial materials contributes to their mode of action [23,24,25,29,30,31,32,33,34,35,36,37].

Just as a range of studies have been devoted to the antimicrobial activity of materials with nanostructured surfaces, other studies have focused on the inherent antimicrobial activity of metals (e.g., Cu, Zn, Ag, Au, Ga, and Fe) [38,39,40]. Before the development of antibiotics, metals were used to treat microbial infections [41], and more recently, metals in the form of nanoparticles were shown to exhibit a broad spectrum of antimicrobial activities due to their small size and high surface-to-volume ratio, which enhance their interaction with microbes [42,43]. For example, nanoparticles of metals (e.g., silver) or of metal oxides (e.g., zinc oxide and copper oxide) that were integrated into carbon-based materials or surfactant-based nano-emulsions were found to be effective in preventing bacterial infections [44,45], water disinfection, and for microbial control [46]. Among these metal-based nanoparticles, copper nanoparticles have attracted considerable attention because of their high redox potential and relatively low production costs [47]. In addition, copper is relatively non-toxic to mammals [48], but shows strong toxicity against a broad range of microorganisms [49,50,51,52]. Hassan et al., for example, reported that copper and copper-based surfaces showed significant antibacterial activity against both Gram-positive and Gram-negative bacteria [53]. It is thus not surprising that copper nanoparticles have already found application in many antimicrobial formulations and products such as biomedical and surgical devices, food processing and packaging, synthetic textiles, and water purification [54,55,56,57].

By virtue of its structure, negative charge, and anion-exchange capacity, polysaccharides have been evaluated as a platform for the complexation with metals such as zinc, copper, and silver with the aim to produce new materials—metal–PS complexes with the synergistic activity of the metal and the polysaccharide [16,17,58,59]. It was shown, for example, that Zn–PS in the form of a hydrogel (with chitosan) exhibited significant antibacterial activity [60]. Similarly, Cu–PS complexes prevented biofilm formation [16,17] and exhibited significant antifungal activity [16].

With the aim of acquiring an in-depth understanding of the role of the spikes protruding from the surfaces of Cu–PS complexes, in their antimicrobial and anti-biofilm activities, we compared a series of Cu–PS complexes prepared from monovalent and divalent copper compounds (Cu_2_O and CuCl vs. CuO and CuCl_2_). These Cu–PS complexes were characterized, and their antimicrobial activities, surface morphologies, and physicochemical properties were compared.

## 2. Results and Discussion

### 2.1. The Cu–PS Complexes: Physicochemical Characteristics

In the first stage, we characterized the various Cu–PS complexes (Table 1). In general, the viscosity of the monovalent Cu–PS complexes was higher than that of the divalent Cu–PS complexes and similar to that of the polysaccharide alone. The conductivity and pH values of all the complexes were higher than the values of the polysaccharide alone. It is worth noting that the zeta potential of the monovalent Cu–PS complexes was lower than that of the divalent Cu–PS complexes (−72 and −65 mV for the monovalent Cu_2_O–PS and CuCl–PS, respectively, compared with −43 and −33 mV for the divalent CuO–PS and CuCl_2_–PS, respectively), but similar to that of the polysaccharide alone (−67 mV), perhaps indicating the higher stability of the monovalent Cu–PS complexes, which was similar to that of the polysaccharide alone.

To examine whether the Cu actually binds to the polysaccharide, the complexes were characterized by FTIR spectroscopy. Figure 1 shows that in both monovalent Cu–PS complexes, a new peak appeared in the transmission spectrum at 1180 cm^−1^, suggesting the formation of a new coordinate bond, as previously shown for the Cu_2_O–PS complex [16]. According to the FTIR spectra, in both monovalent complexes, Cu ions form coordinate bonds with the polysaccharide, whereas in the divalent Cu–PS complexes, the Cu ions probably interact electrostatically with the polysaccharide. In all of the spectra, the broad band centered at 3260 cm^−1^ was assigned to O−H stretching vibrations, and the weak signal at 2926 cm^−1^ to C−H stretching vibrations [61]. The broad band at around 1220−1260 cm^−1^, exhibited by all the samples, was assigned to sulfated ester groups (S=O), which are a characteristic component of the sulfated polysaccharides of red microalgae and seaweeds [61,62,63,64].

In accordance with Figure 1 showing Cu binding to the polysaccharide, the copper release profiles of the Cu–PS complexes are presented in Figure 2. The highest Cu release from the complexes into distilled water was detected for the divalent Cu–PS complexes (CuO–PS and CuCl_2_–PS), whereas there was no substantial release of copper from the monovalent Cu–PS complexes (Cu_2_O–PS and CuCl–PS): The copper release from the divalent Cu–PS complexes was about 100 times that from the monovalent Cu–PS complexes.

SEM micrographs and EDS analysis provided information on the surface morphology of the various Cu–PS complexes (Appendix A). It can be seen that the monovalent Cu–PS complexes (Appendix A) have a larger pore size of up to 50 µm in diameter vs. the 10-µm diameter pores of the divalent Cu–PS complexes (Appendix A). An EDS analysis (Appendix A) showed that the Cu was equally distributed in all the Cu–PS complexes and that they all exhibited the same main peaks of sulfur, carbon, and oxygen.

### 2.2. Antimicrobial and Antibiofilm Activities of Monovalent vs. Divalent Cu–PS Complexes

Our previous findings that the Cu_2_O–PS complex exhibits significant antimicrobial activity, exceeding that of the polysaccharide alone or of Cu_2_O alone, were attributed to the spikes induced on the surface of the Cu_2_O–PS complex [16]. To obtain a comprehensive insight into this significant antimicrobial activity, we compared monovalent vs. divalent Cu–PS complexes in terms of growth inhibition and cell viability (Figure 3A). The two monovalent Cu–PS complexes were markedly more inhibitory than the divalent Cu–PS complexes for the fungus *C. albicans* and for different Gram-negative (*A. baumannii*, *P. aeruginosa*, and *E. coli*) and Gram-positive (*S. aureus* and *B. subtilis*) bacteria: both monovalent Cu–PS complexes almost completely inhibited the growth of *C. albicans* (93 and 89% inhibition for Cu_2_O–PS and CuCl–PS, respectively) vs. untreated cells, whereas the divalent Cu–PS complexes and the *Porphyridium* sp. polysaccharide alone caused moderate growth inhibition of *C. albicans*. For all of the bacterial species, the monovalent Cu–PS complexes caused 75–83% inhibition of growth. Copper salts alone did not inhibit the growth of the fungus or of any of the bacterial species (Appendix A).

To investigate the effect of the Cu–PS complexes on cell viability, the lysis of the fungal and bacterial cells was evaluated using the colony forming unit (CFU) assay after 24 h of incubation of the microbial cells with the complexes (Figure 1B). The results showed that the two monovalent Cu–PS complexes exhibited the most effective activity against *C. albicans* (i.e., 5–12% viable cells remained after treatment compared with 55–90% viable cells after exposure to the polysaccharide or Cu salts alone). The divalent Cu–PS complexes showed moderate activity against all of the tested microorganisms (~21–42% viability).

Taken together, these two experiments indicate that the monovalent Cu–PS complexes were superior to the divalent Cu–PS complexes in terms of both inhibiting growth and reducing cell viability (leading to cell death) (Figure 3; Appendix A). Of the two monovalent Cu–PS complexes, the effects of the Cu_2_O–PS complex were more marked than those of the CuCl–PS complex. As shown previously, the inhibition of the growth of *C. albicans* was significantly higher than that of the bacteria (about 1.5–7 times).

Additional support for the ability of the monovalent (vs. divalent) Cu–PS complexes to inhibit bacterial growth was obtained by evaluating the relative swarming motility in the presence of the complexes of a pathogenic nosocomial infectious Gram-negative bacterium *P. aeruginosa* PA14 as a model microorganism. For this purpose, the bacteria were cultured on semisolid agar plates, each layered with a different Cu–PS complex (Figure 4). The monovalent Cu–PS complexes significantly decreased the swarming ability of *P. aeruginosa* PA14 (Figure 4, top row), resulting in an approximately 5-fold decrease in surface coverage compared to the polysaccharide alone. The Cu compounds did not have any effect on the swarming ability of the bacterium (Figure 4, bottom row).

*P. aeruginosa* PA14 was also used as a model microorganism to study the effect of the Cu–PS complexes on the development of dynamic biofilms. The effect of the Cu–PS complexes on the growth of *P. aeruginosa* was evaluated in a microfluidic system for biofilm formation [65,66,67] (Figure 5). CLSM analysis revealed that after 48 h, thick, dense biofilms of *P. aeruginosa* PA14 developed, which uniformly covered the untreated glass and the polysaccharide-treated surfaces (Figure 5A,B). The monovalent Cu–PS complexes almost completely prevented bacterial adherence (Figure 5C,D), whereas the divalent Cu–PS complexes had only a partial effect (Figure 5E,F). Quantitative analysis of the biofilm thickness (with the Imaris MeasurementPro) showed that the monovalent Cu_2_O–PS and CuCl–PS complexes exhibited 98% and 88% of biofilm formation inhibition, respectively, compared with ‘No treatment’ (Figure 5G). In addition, a calculation (from Figure 5G) of the ratio of dead/live cells clearly showed that the effect of the Cu_2_O–PS complex was 10 times greater than that of the other monovalent complex (CuCl-PS) and 12.5–20 times greater than that of the polysaccharide alone or the divalent Cu–PS complexes, indicating the lethal effect of the Cu_2_O–PS complex (Figure 5H). It is possible that the spikes that protrude from the surface of the Cu_2_O–PS complex interfere with the initial stage of biofilm formation, thereby preventing further biofilm development.

### 2.3. Surface Topography of the Cu-PS Complexes: Spike Nanotopography

Since we hypothesized that the antimicrobial and antibiofilm activities of the Cu–PS complexes derived from spikes on their surfaces, we used AFM to further study the spike structure. The surface topography and 3D images of the complexes are shown in Figure 6A, and an analysis of the spike parameters, using Gwyddion and ImageJ software, is presented in Figure 6B. It can be seen that the spikes of the monovalent Cu–PS complexes were significantly higher than those of the divalent Cu–PS complexes and those of the polysaccharide itself (1000 nm vs. 24 nm). The density (spikes/µm^2^) of the spikes of the monovalent Cu_2_O–PS complex was 2.5 times greater than that of the monovalent CuCl–PS complex and 5 times greater than that of both divalent Cu–PS complexes. However, the spike thickness was the lowest in the monovalent Cu_2_O–PS complex (20 nm), with the spikes of the divalent Cu–PS complexes being 4.5–5 times thicker (50–90 nm). Spike height thus appeared to be positively correlated with density and negatively correlated with thickness. A schematic illustration of the spikes is presented in the Appendix A (Appendix A). An additional parameter, the spike aspect ratio, was used to characterize the spike structure (Figure 6B). The aspect ratio of the Cu_2_O–PS complex was 2.5 times higher than that of the CuCl–PS complex and 100–200 times higher than that of the other complexes. The roughness (Rq) as obtained from the AFM scans of the Cu_2_O–PS (~1040.0 nm) and CuCl–PS (~874.7 nm) complexes was significantly greater than that of the divalent Cu–PS (~57.6–75.3 nm) complexes or that of the polysaccharide per se (~11.2 nm). To further understand the effect of spikes on the antimicrobial activity, we ‘neutralized’ the effect of spikes from the surface chemistry of the Cu–PS complexes by using a gold coating. In this way, we could separate the effect of the surface (not coated) from that of the spikes. For the monovalent Cu–PS complexes, the spike heights did not change (remained 1000 nm) upon the gold coating (Figure 6B), whereas the thickness increased, and the density (spikes/1 µm^2^) decreased compared with the non-coated surfaces. A similar trend was observed for the divalent Cu–PS complexes.

When the surfaces of the Cu–PS complexes were coated with gold (Figure 7), more bacterial cells were observed vs. the non-coated surfaces, emphasizing the influence of the size and sharpness of the spikes on the antimicrobial activity of the complexes. The coated and uncoated surfaces showed similar patterns of biofilm formation. Nonetheless, the inhibitory effect of Cu_2_O–PS was significantly different from that for the other complexes, probably indicating the effect of the spikes above and beyond that of the surface chemistry. In general, the areas covered by *P. aeruginosa* PA14 cells (%) were similar for the non-coated and gold coated complexes (Pearson correlation *r* = 0.98) (Appendix A).

### 2.4. Effect of Spikes on Perforation of Bacterial Membranes

An additional approach was taken to examine whether the spikes did, in fact, pierce the microbial membrane, by using a bioluminescence assay. For this purpose, *E. coli* TV1061 was used as a model. When these bacterial cells are incubated with certain toxicants (e.g., ethanol), bioluminescence is generated once the lux operon has been transcribed and translated to produce luciferase, an event triggered by the adjacent promoter, which is sensitive to certain toxicants that can damage the cell membrane [68]. The *E. coli* TV1061 reporter strain was incubated with the various Cu–PS complexes, and the luminescence resulting from luciferase leakage was determined. As can be seen from Figure 8, the cytotoxic effect of the two monovalent Cu–PS complexes was superior to that of the divalent Cu–PS complexes (the polysaccharide itself did not have any effect), with the bioluminescence for the Cu_2_O–PS complex being similar to that of the positive control (20% ethanol, EtOH). The area under the peak was used as a measure of bioluminescence accumulation resulting from membrane perforation and cytoplasmic leakage (Figure 8B). Scanning electron micrographs of the damaged cells are presented in Appendix A.

## 3. Materials and Methods

### 3.1. Algal Growth and Polysaccharide Production

*Porphyridium* sp. (UTEX 637) from the culture collection of the University of Texas at Austin was grown in artificial seawater [69]. Algal cultivation and polysaccharide production were performed as previously described [70]. Briefly, the cultures were grown in polyethylene sleeves and illuminated continuously from the side with fluorescent cool-white lamps at an irradiance of 150 μE m^−2^ s^−1^ and aerated with sterile air containing 3% CO_2_. Cells were harvested at the stationary phase of growth by continuous centrifugation (CEPA, Carl Padberg Zentrifugenbau GmbH, Lahr, Germany). The supernatant containing the dissolved polysaccharide was collected and filtered using crossflow filtration to remove salts and metabolites (MaxCell™ hollow fiber microfiltration cartridge, pore size 0.45 μm, membrane area 2.5 m^2^) and concentrated to 0.7% (*w/v*) polysaccharide. The resulting polysaccharide was sterilized in an autoclave and stored at 4 °C. The sugar content was determined using the phenol-sulfuric acid reaction [71].

### 3.2. Preparation of Cu–PS Complexes

The various Cu–PS complexes were prepared by adding the copper salts (Cu_2_O and CuO from Fisher Scientific, Loughborough, UK; CuCl_2_ and CuCl from Sigma–Aldrich, Brøndby, Denmark) to 20 mL of a 0.7% (*w/v*) polysaccharide solution to a final copper concentration of 500 ppm. The Cu–PS complexes were stirred gently with a magnetic stirrer for 24 h at room temperature. Copper concentration in the Cu–PS complexes was determined by inductively coupled plasma optical emission spectrometry (SPECTRO ARCOS ICP-OES analyzer, SPECTRO Analytical Instruments GmbH, Kleve, Germany).

### 3.3. Physicochemical Characterization of Porphyridium sp. Polysaccharide Solutions

The viscosity of the polysaccharide solutions was determined with a Brookfield digital viscometer, 30 rpm at room temperature with a 31 cylindrical spindle (Brookfield AMETEK SC4-31, Middleboro, Massachusetts, USA). The conductivity and pH of the polysaccharide solutions were determined with a pH/mV/Cond./TDS/Temp. meter 86,505 (AZ Instrument Corp., Taichung City, Taiwan) at room temperature. Zeta potential (electrokinetic potential) was measured using a Zetasizer Nano ZS (Malvern Instruments Ltd., Malvern, UK). Data were analyzed using the Smoluchowski model.

### 3.4. Fourier Transform Infrared (FTIR) Spectroscopy

Spectra were obtained on an iN-10 FTIR Thermo Fisher Microscope spectrometer equipped with a narrow-band liquid-nitrogen-cooled MCTA detector. Samples of the Cu–PS complexes and the polysaccharide alone were lyophilized at −55 °C for 24 h in 96-well plates. The spectra were recorded in three areas per sample in the range of 4000 to 650 cm^−1^, at 2 cm^−1^ resolution and 64 scans; area of detection was 25 × 25 mm. The FTIR data were collected using OMNIC Picta software (Thermo Scientific OMNIC Series Software, Madison, WI, USA). Automatic baseline correction was used.

### 3.5. Copper Release from the Cu–PS Complexes

The mode of copper leaching from the Cu–PS complexes was monitored using cellulose tubular membranes (nominal MWCO: 12,000–14,000, Merck) containing the Cu–PS complexes against distilled water with constant shaking (100 rpm) at 37 °C. The released copper ions were quantified after 72 h of incubation. The copper content in the distilled water was determined by ICP-OES.

### 3.6. Scanning Electron Microscopy/Energy Dispersive X-ray Spectroscopy (SEM-EDS) Analysis

To determine the morphological properties of the Cu–PS complexes and the polysaccharide, SEM images were obtained on an FEI ESEM Quanta 200 instrument at an accelerating voltage of 20 kV and analyzed by EDS (FELMI-ZFE, Graz, Austria). Samples were prepared for SEM scanning as follows. Freeze-dried Cu–PS complexes were attached to specimen holders with double-sided carbon tape and then sputtered with gold. The pore sizes of the complexes in the SEM images were measured with ImageJ software. Three images were randomly selected from each sample, and 30 measurements of pore sizes were acquired from each image.

### 3.7. Microbial Cultures and Growth Conditions

The antimicrobial activities of the Cu–PS complexes were determined for a variety of model microorganisms cultured as follows. *Acinetobacter baumannii* (ATCC No. 43498, USA), *E. coli* (ATCC No. 1100101, USA), and *Pseudomonas aeruginosa* PA14 (ATCC No. 109246, Manassas, VA, USA) were grown for 24 h in LB Broth (Miller) (Sigma-Aldrich); *Staphylococcus aureus* (ATCC No. 33591, USA) was cultured for 24 h in BD™ Tryptic Soy Broth (TSB; soybean-casein digest medium); and *Bacillus subtilis* (ATCC No. 19659, USA) was grown for 24 h in LB medium (Difco Luria-Bertani medium, Lennox). *Candida albicans* (ATCC No. 10231, USA, supplied by the Clinical Microbiology Laboratory of Dr. Yossi Paitan, Meir Medical Center, Kfar Saba, Israel) was cultivated in potato dextrose broth (PDB; HiMedia) for 48 h with shaking (120 rpm) at a constant temperature of 37 °C. For the antibiofilm study, *P. aeruginosa* PA14 was inoculated for 24 h into AB trace Minimal Medium, supplemented with 30 mm glucose at a constant temperature of 37 °C.

### 3.8. Microbial Growth Inhibition

The antimicrobial activities of the Cu–PS complexes and of the . polysaccharide alone against *A. baumannii*, *P. aeruginosa* PA14, *E. coli*, *S. aureus*, *B. subtilis*, and *C. albicans* were examined by following the growth curves and viability of the microorganisms. To evaluate the growth curves for the different species of bacteria and the fungus, 100 µL of a solution of the polysaccharide (0.7% *w/v*), a Cu–PS complex (0.7% *w/v* polysaccharide with 500 ppm of copper ions) or copper ions (500 ppm) were mixed with 900 µL of the relevant medium and 10 µL of microbial culture at OD = 1. A sample of 200 µL of each combination was incubated with shaking in 96-well plates at 37 °C for 14 h for *A. baumannii*, *P. aeruginosa* PA14, *E. coli*, *S. aureus*, and *B. subtilis* or 48 h for *C. albicans*. The turbidity of the medium was measured hourly with a micro-plate reader (BioTek Instruments, Santa Clara, CA, USA) at a wavelength of 600 nm. Growth inhibition was calculated from the following formula:(1)$$Inhibitory\;index\;\left(\%\right)=\left(1-\frac{OD_{treatment}-OD_{treatment\_blank}}{OD_{control}-OD_{control\_blank}}\right)\times 100$$
where *OD_treatment_* is the absorbance of the sample of each Cu–PS complex or polysaccharide plus bacteria at t = 14 h (in the logarithmic phase of growth) or fungus at t = 48 h; *OD_treatment_blank_* is the absorbance of the same sample without bacteria or fungus at the same time; *OD_control_* is the absorbance of LB or TSB (as above-mentioned for each bacterial species) plus bacteria or PDB plus fungus; and *OD_control_blank_* is the absorbance of the same sample without bacteria or fungus at the same time.

For the determination of cell viability, cells were incubated in test tubes containing 900 µL of LB, TSB, or PDB, as relevant, mixed with 100 µL of each Cu–PS complex solution (0.7% *w/v* polysaccharide with 500 ppm copper ions) vs. their respective controls (LB, TSB, or PDB alone, or solutions of polysaccharide or copper ions). After 24 h, 100 µL of each sample was plated on a LB, TSB, or PDB agar plate after serial dilution and incubated overnight at 37 °C. The following morning, the colony-forming units (CFUs) were counted.

### 3.9. Bacterial Surface Translocation

*P. aeruginosa* PA14 was grown overnight in M9 medium [62 mm potassium phosphate buffer (pH 7), 7 mm (NH_4_)_2_SO_4_, 2 mm MgSO_4_, 10 μm FeSO_4_, 0.4% (*w/v*) glucose, 0.5% Casamino Acids)]. The cultures were then diluted 1:10 into fresh M9 medium and cultured to the mid-log phase (i.e., OD_600_ of 0.4 to 0.6). Inoculate of 1–2 μL were placed in the middle of the swarming plates (M9 solidified with 0.5% (*w/v*) Difco agar and 0.1% of a Cu–PS complex) to enable the assessment of the surface coverage after 24 h of growth at 37 °C.

### 3.10. Confocal Laser Scanning Microscopy (CLSM)

A continuous-culture flow cell [66], with three channel μ-slide sizes of 17 × 3.8 × 0.4 mm (ibid), was used as a solid platform for biofilm growth. The flow cell system was inoculated with a 0.05 OD_600_ dilution of *P. aeruginosa* PA14 culture. The flow was initiated after 1 h at a flow rate 10 mL/h of AB trace Minimal Medium, supplemented with 30 mm glucose. The system was incubated at 37 °C for 24 h. Biofilms were visualized by CLSM, and the 3D visualization was processed using IMARIS software. Data acquisition and processing of bacterial biofilms were performed by an FV1000 (Olympus, Tokyo, Japan) CLSM equipped with a 0 × 1.35 NA lens. Bacteria were pre-stained using a LIVE/DEAD BacLight Viability Staining Kit (Molecular Probes Inc., Eugene, OR, USA). A stock solution of SYTO™ 9 and propidium iodide (PI) stain were diluted in doubly distilled water to a concentration of 1.5 μL/mL and injected into the flow channels. SYTO 9 stained the live cells (green), and PI stained the dead cells (red). The excitation wavelength for SYTO 9 was 488 nm and the emission wavelength was 515 nm, while for PI, excitation and emission were recorded at 530 nm and 617 nm, respectively. The collected images were processed using 3D reconstruction IMARIS software (Bitplane AG, Zürich, Switzerland). Biofilm thickness (μm^3^/μm^2^) was calculated over a given sampling of slices (~320 μm × 320 μm).

### 3.11. Atomic Force Microscopy (AFM)

Glass coverslips were immersed in 2.5 M HCl for 10 min and then rinsed first with ethanol (99.9%) and then with distilled water. Thereafter, 10 µL of the relevant Cu–PS complex or of the polysaccharide itself was applied to the glass surface and dehydrated by autoclaving at 121 °C for 40 min. Topographical images of the pre-dried Cu–PS complexes and that of the polysaccharide were acquired on a Dimension-3100 microscope (Bruker). Samples were imaged at a scan rate of 0.5−1 Hz with a 512 × 512 pixel resolution in tapping mode. Several scans were carried out over a given surface area (~0.65 μm × 0.67 μm). Electrostatic potential distribution measurements were conducted with a scanning Kelvin probe microscopy (SKPM) using an MFP-3D-Bio inverted optical microscope system with an ARC2 controller (Asylum Research, Oxford Instruments). The images were recorded by SKPM in NAP mode, with a two-pass method. On the first pass, the cantilever was oscillated mechanically, and the surface topography was recorded, while in the second pass, during which AC and DC voltage were applied, the surface potential was recorded. AFM images were analyzed manually using Gwyddion and ImageJ software. Three images were randomly selected for each sample, and 30 measurements (maximal spike height, spike thickness, and density) were acquired from each image. The aspect ratio was then calculated from the data collected from the AFM images.

### 3.12. Antimicrobial and Antibiofilm Activities of Gold-Coated Cu–PS Complexes

Glass coverslips were immersed in 2.5 M HCl for 10 min and then rinsed with ethanol (99.9%) and distilled water. Thereafter, 10 µL of a Cu–PS complex or of the polysaccharide were applied on the glass surface and dehydrated by autoclaving (121 °C for 40 min). The glass slides were then attached to specimen holders with double-sided carbon tape and coated with a 1-nm layer of gold using an EMITECH K575x sputtering device (Emitech Ltd., UK). Strain PA14 of *P. aeruginosa* was exposed to the gold coated and uncoated surfaces, and the morphological properties of the coatings and the biofilm structures were analyzed using high-resolution scanning electron microscopy (HR-SEM) after 24 h of incubation. The samples were prepared for SEM studies as follows. After fixation in 2.5% buffered glutaraldehyde, the samples were dehydrated with an increasing serial ethanol gradient (25%, 50%, 75%, 90%, 95%, and 100%) and immersed in hexamethyldisilazane (HMDS)/ethanol gradient solutions (1:3, 1:1, 3:1). The treated specimens were air dried for 4 h, and in preparation for SEM scanning (JSM-7400F, JEOL), they were sputter coated with a 20-nm layer of gold, as described above. SEM images were analyzed using ilastik and ImageJ software.

### 3.13. Bioluminescence Accumulation Assay

For this assay, we used the bioluminescence reporter strain *E. coli* TV1061, which is sensitive to general cytotoxic damage including the cell membrane [72]. This bacterial strain harbors a fusion of the luxCDABE reporter gene and the promoter for the heat-shock gene grpE (provided by S. Belkin, The Hebrew University of Jerusalem, Israel). Prior to the exposure of the *E. coli* TV1061 cells to the Cu–PS complexes, the bacterial cells were cultivated in 10 mL of LB medium (10 g/L tryptone; 5 g/L yeast extract; 5 g/L NaCl) overnight at 37 °C in a shaking incubator at 120 rpm (NB-205LF,N-BIOTEK, SciMed (Asia) Pte Ltd., Singapore). Cultures were then diluted to approximately 10^7^cells/mL and regrown in 10 mL of LB at 30 °C without shaking, until the early exponential phase (i.e., OD_600_ = 0.2), as determined by a UVmini-1240, UV–VIS spectrophotometer (Shimadzu, Singapore). Bioluminescence was measured using a Luminoskan Ascent Luminometer (ThermoFisher Scientific, Waltham, MA, USA). Measurements were performed in white 96-well microtiter plates (NUNC) containing 90 μL of the bacterial culture at OD_600_ = 0.2. The treatments (10 μL in different concentrations) were added to each well (n = 3 for each treatment). During the measurements (10 h), temperature was maintained at 26 °C, and the plates were continuously shaken. The negative control was obtained by adding 10 μL of LB to the bacterial culture and the positive control was obtained by the addition of 2% (*v/v*) ethanol. Luminescence values are presented in relative light units (RLU). Area under the peak was calculated according to the following formula:(2)$$Area\;under\;peak=\int_{0}^{600}f\left(x\right)dx$$
where *f*(*x*) is the equation plot of each treatment.

## 4. Conclusions

The involvement of various surface nanostructures in the antimicrobial activities of different microorganisms has been studied [22,23,24,25,29,30,31,32,33,34,36,38]. We have previously suggested that the antimicrobial activities of the Cu_2_O–polysaccharide complex are derived from the spikes produced on the surface of the complex, which penetrate and hence disrupt the microbial membrane, thereby causing cell death [16]. With the aim to elucidate the role of the spikes in the antimicrobial activities of Cu–PS complexes, we sought to understand how the structure of the spikes (size, height, width, and density) affected the antimicrobial activities of the complexes. Two important findings were that (i) the roughness—used as an indicator of surface topography—was markedly greater (~15 fold) in the monovalent Cu–PS complexes than in the divalent complexes, and (ii) the spike aspect ratio—reflecting the antimicrobial effectiveness of the spikes –was significantly higher for the two monovalent Cu–PS complexes (Cu_2_O–PS and CuCl–PS) than for the divalent Cu–PS complexes and for the polysaccharide per se (Figure 9A). We therefore posited that the sharp spikes of the monovalent Cu–PS complexes exerted their antimicrobial action by perforating the microbial membrane. This was supported by the enhanced accumulation of bioluminescence due to cytoplasm spilling through the pierced cell membrane of Cu–PS complex-treated bacteria (Figure 9B). Moreover, the highest growth inhibition was directly correlated with the highest bioluminescence (Figure 9C) for the highest aspect ratio of 50 (for the Cu_2_O-PS complex). Taken together, these results (spike height, cytoplasm spilling, and growth inhibition), in addition to the antibiofilm activity (Figure 5), clearly point to a direct effect of the spikes—via membrane disruption—on the bacteria, which culminated in cell death (Figure 9). The generality of the phenomenon is emphasized since various bacteria were used (Figure 9C and Appendix A).

The size of the aspect ratio was directly correlated with microbial death for all the microorganisms studied, with the highest growth inhibition and the lowest cell viability being found for the fungus *C. albicans*. It would appear that the superior antimicrobial activity of the monovalent Cu–PS complexes mainly stems from their surface topography, namely, the presence of sharp spikes, since neutralizing the effect of surface chemistry (by coating the surfaces with a thin layer of gold) only led to a slight reduction in the killing efficiency (Figure 7). It is worth adding here that the stability of the monovalent Cu–PS complexes is supported by the covalent bonding of the Cu to the polysaccharide molecule (Figure 1) and the absence of leakage of Cu from the complexes (Figure 2).

The results of this study can serve as the basis for the production of new functional metal–polysaccharide complexes with a wide range of applications.

## Figures and Tables

**Figure 1 marinedrugs-20-00787-f001:**
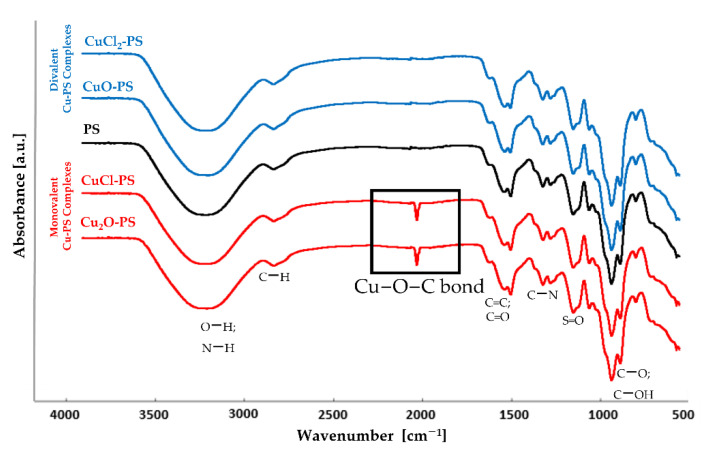
FTIR transmission spectra of the divalent Cu–PS complexes (blue), the monovalent Cu–PS complexes (red) and the polysaccharide (black). All Cu–PS complexes contained 0.7% polysaccharide (*w/v*) and 500 ppm copper. To facilitate ease of viewing, the spectra are displaced with respect to the Y axis.

**Figure 2 marinedrugs-20-00787-f002:**
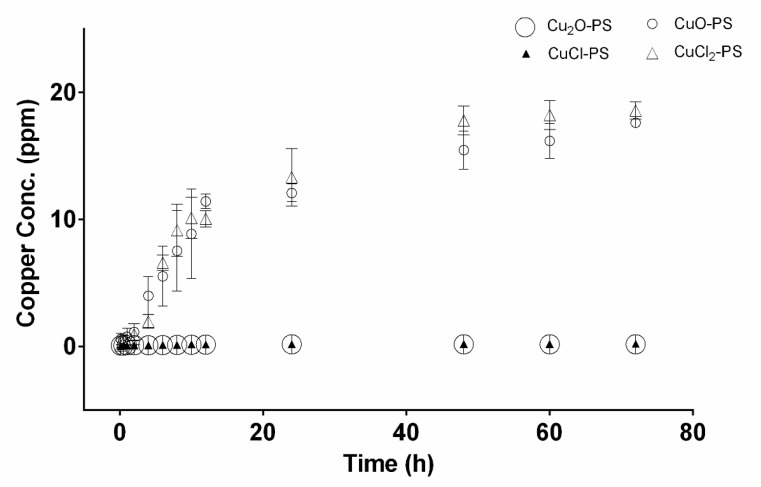
Copper-release profiles from monovalent and divalent Cu–PS complexes. The copper concentration was determined in distilled water. Data represent the average values of three independent experiments. All the Cu–PS complexes contained 0.7% polysaccharide (*w/v*) and 500 ppm copper. Copper concentration was measured using a SPECTRO ARCOS ICP-OES analyzer.

**Figure 3 marinedrugs-20-00787-f003:**
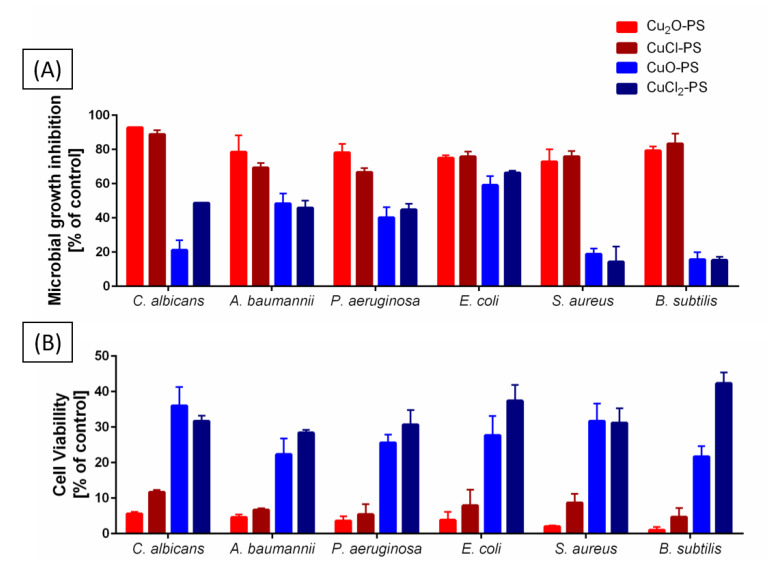
Effect of the monovalent and divalent Cu–PS complexes on (**A**) the inhibition of growth and (**B**) the cell viability of a fungus (*Candida albicans*), Gram-negative bacteria (*Acinetobacter baumannii*, *Pseudomonas aeruginosa*, and *Escherichia coli*), and Gram-positive bacteria (*Staphylococcus aureus* and *Bacillus subtilis*). All the Cu−PS complexes contained 0.07% (*w/v*) polysaccharide and 30 ppm copper. For the growth inhibition experiments, the control was the absorbance of the relevant growth medium with only the bacterium or the fungus (see the experimental section). The microbial cultures were incubated with shaking in 96-well plates at 37 °C for 14 h for each bacterial species or 48 h for *C. albicans*. Each sample was plated on an agar plate of the relevant medium after serial dilution and incubated overnight at 37 °C. CFUs were counted the following morning and were assessed vs. untreated cells. Values are the means ± standard error of mean (SEM)of three independent experiments performed in triplicate. All of the results were significantly different from their relative controls (ANOVA; *p* < 0.05).

**Figure 4 marinedrugs-20-00787-f004:**
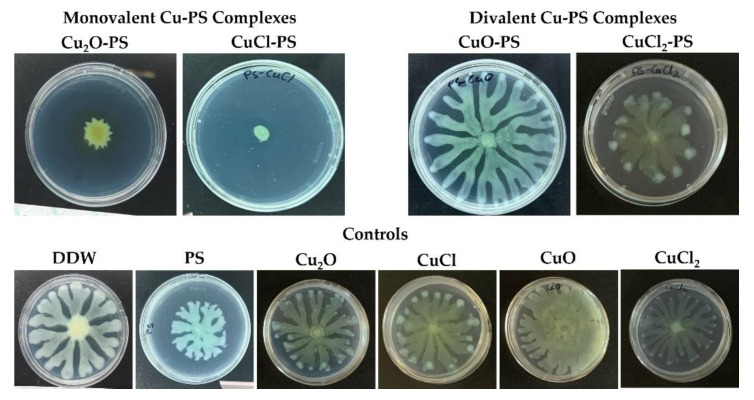
Effect of monovalent and divalent Cu–PS complexes on the swarming motility of *P. aeruginosa* PA14. Top row: Motility of *P. aeruginosa* PA14 in the presence of Cu–PS complexes. Bottom row: Controls. The bacteria were inoculated into the center of each plate consisting of M9 solidified with 0.5% (*w/v*) Difco agar and containing 0.1% of the relevant Cu–PS complex. Surface coverage was assessed after 24 h of growth at 37 °C. All of the Cu–PS complexes contained 0.7% polysaccharide (*w/v*) and 500 ppm copper. For the control treatments, the copper concentration in the copper-containing plates was 500 ppm, and the PS plate contained 0.7% *Porphyridium* sp. polysaccharide (*w/v*).

**Figure 5 marinedrugs-20-00787-f005:**
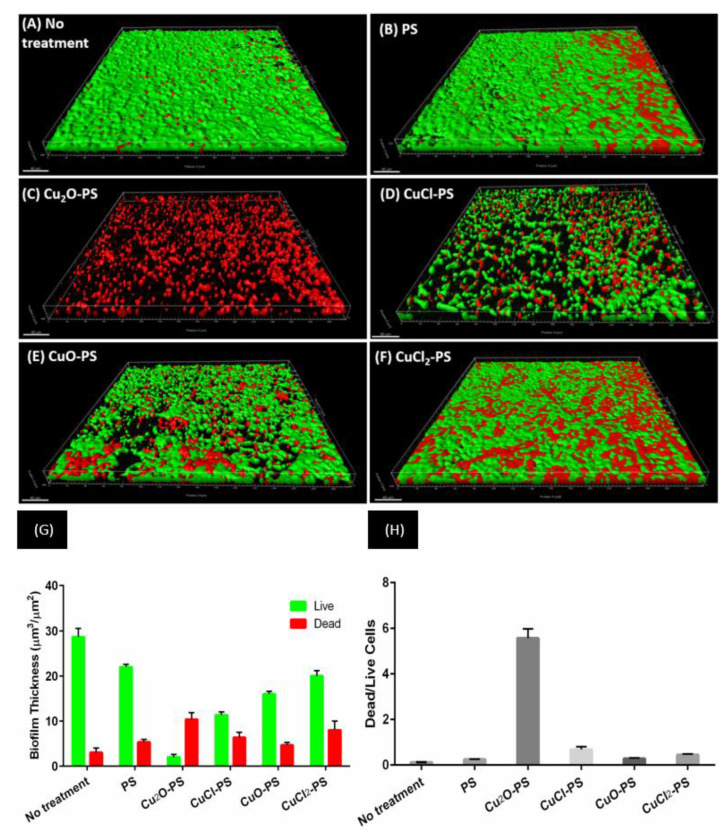
Effect of monovalent and divalent Cu–PS complexes on *P. aeruginosa* PA14 biofilm formation. Forty-eight hours after inoculation, biofilm formation was assessed by CLSM on (**A**) an untreated surface, (**B**) a surface pre-coated with *Porphyridium* sp. polysaccharide, (**C**,**D**) surfaces pre-coated with monovalent Cu–PS complexes, and (**E**,**F**) surfaces pre-coated with divalent Cu–PS complexes. Viable cells stained green, and dead cells stained red with the BacLight^®^ DEAD/LIVE Kit for scanned areas of ~318 μm × 318 μm. (**G**) Cell-layer thickness (μm^3^/μm^2^) for live and dead cells; values are means ± standard error of mean (SEM). The results are presented for three independent sets of flow-cell experiments, each containing 30 measurements. Significant differences between the groups (*p* < 0.0001) by two-way ANOVA, followed by Tukey’s test. (**H**) Ratio of dead-to-live cells (calculated from Figure 5G). All of the Cu–PS complexes contained 0.7% polysaccharide (*w/v*) and 500 ppm copper.

**Figure 6 marinedrugs-20-00787-f006:**
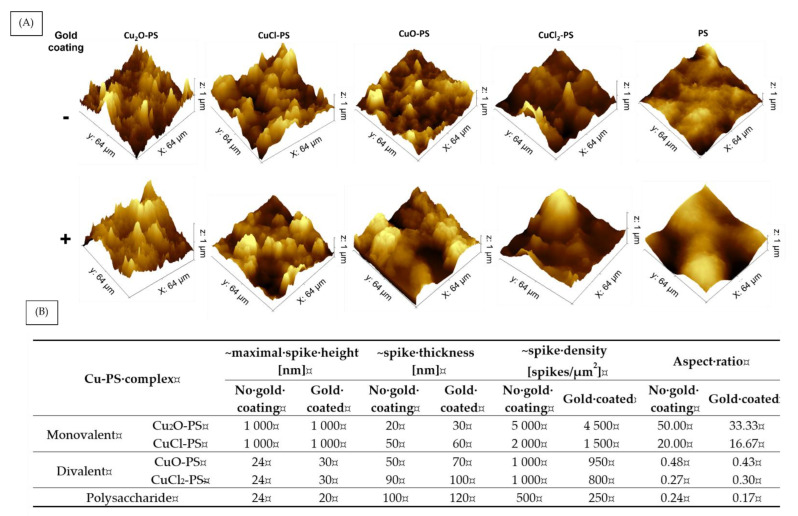
Surface topography and spike distribution of monovalent and divalent Cu–PS complexes with and without the gold coating. (**A**) AFM 3D images and (**B**) spike parameters. 3D images of the Cu–PS complexes and the polysaccharide were manually analyzed using Gwyddion and ImageJ software. Spike thickness and density were calculated manually using ImageJ and the aspect ratio was calculated from the data collected from the AFM images analyzed by Gwyddion and ImageJ software. All of the Cu–PS complexes contained 0.7% polysaccharide (*w/v*) and 500 ppm copper.

**Figure 7 marinedrugs-20-00787-f007:**
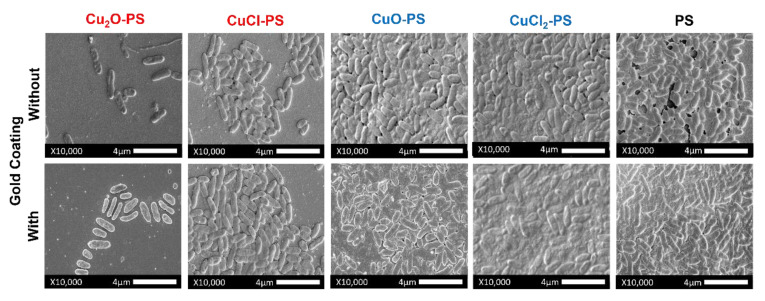
SEM micrographs showing the effect of monovalent and divalent Cu–PS complexes that were surface coated with gold vs. non-coated on *P. aeruginosa* PA14 biofilm formation. ×10,000, scale bar = 4 µm. All of the Cu–PS complexes contained 0.7% polysaccharide (*w/v*) and 500 ppm copper. Images of the control (glass surface alone) are presented in Appendix A.

**Figure 8 marinedrugs-20-00787-f008:**
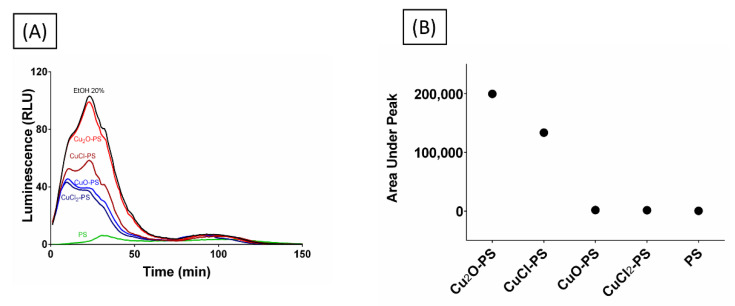
(**A**) Bioluminescence signal from *E. coli* TV1061 induced by monovalent and divalent Cu–PS complexes. The bioluminescence was measured in relative light units (RLU) at 490 nm (i.e., the wavelength attributed to bacterial luciferase). (**B**) Area under the peak showing the leakage of luciferase from the bacterial cells. All of the Cu−PS complexes contained 0.07% (w/v) polysaccharide and 30 ppm copper. In the copper salts (Cu_2_O, CuCl, CuO, CuCl_2_), the same pattern of bioluminescence as the PS alone was shown (not shown).

**Figure 9 marinedrugs-20-00787-f009:**
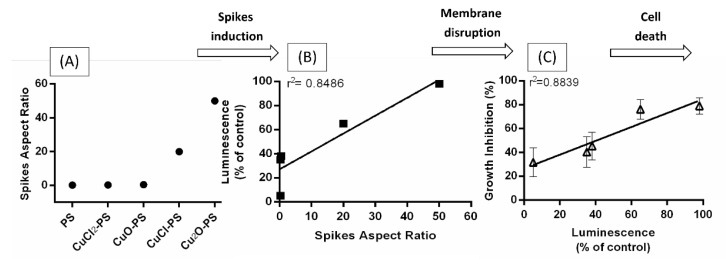
Microbial cell death as a function of the characteristics of the spikes induced on the Cu–PS complexes. (**A**) Aspect ratios (•) of the spikes of the various Cu–PS complexes. (**B**) Luminescence of *E. coli* TV1061 as a function of the aspect ratio (■). (**C**) Microbial growth inhibition as function luminescence (∆) for five different microorganisms: *Candida albicans, Acinetobacter baumannii*, *Pseudomonas aeruginosa*, *Escherichia coli*, *Staphylococcus aureus*, and *Bacillus subtilis*. All the Cu−PS complexes contained 0.07% (*w/v*) polysaccharide and 30 ppm copper. The results presented are calculated from Figure 3 and Figure 6.

**Table 1 marinedrugs-20-00787-t001:** Physicochemical characteristics of the Cu–PS complexes.

	Monovalent Cu–PS Complexes		Divalent Cu–PS Complexes		Control
	Cu_2_O–PS	CuCl–PS	CuO–PS	CuCl_2_–PS	Polysaccharide
Viscosity [cP]	2415.3 ± 60.5	2230.0 ± 72.1	1978.3 ± 24.8	1979.0 ± 18.3	1733.7 ± 129.8
Conductivity [µS]	3303.0 ± 265.6	2930.7 ± 122.2	1943.3 ± 50.3	1230.0 ± 81.9	1043.3 ± 92.9
ζ-potential [mV]	−72.5 ± 1.3	−65.7 ± 4.1	−43.6 ± 0.5	−33.9 ± 0.3	−67.5 ± 2.3
pH	6.3 ± 0.7	6.4 ± 0.9	6.1 ± 0.4	6.0 ± 0.2	5.0 ± 0.4

All the Cu–PS complexes contained 0.7% polysaccharide (*w/v*) and 500 ppm copper. Data are the means ± standard error of triplicate samples. Details of the analyses are given in the Materials and Methods section.

## Data Availability

Not applicable.

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
