# Peer review of "Complexes of Cu–Polysaccharide of a Marine Red Microalga Produce Spikes with Antimicrobial Activity"

_marinedrugs, 2022, doi:10.3390/md20120787_

Round 1
Reviewer 1 Report
Interesting paper. good work. should be presented better e.g. the quality of figures. All what is shown must be readable and must be clearly defined (line colours etc)
There are 1000nm spikes. And 5000 spikes per µm2. Show a sketch of the algae membranes with the spikes in a transect. For all used polysacharids; and for the harvested complexes
Fig 1 green trace nearly undistinguishable to the blue. Differentiate clearly.
Show SEM graphs of membranes and complexes
Fig 2. Clearly differentiate bottom line. It is an unclear mixture
Fig 4 define the controls
Fig 6 A too small to recognize bars and details
Conclusion is far too long
Supporting information: Draw a scetch of SEM micrographs in Figure S1
It is hard to believe to see 5000spike / µM. Do you mean mm?
Author Response
Reviewer #1
- Interesting paper. good work. should be presented better e.g., the quality of figures. All what is shown must be readable and must be clearly defined (line colors etc.)
Author Response: We have modified the figures as can be seen in the revised manuscript- track changes.
- There are 1000nm spikes. And 5000 spikes per µm2. Show a sketch of the algae membranes with the spikes in a transect. For all used polysaccharides; and for the harvested complexes
Author Response: We would like to clarify: There are no algae and no algae membranes here. The polysaccharide is released from the cell wall of the algae. We have collected the polysaccharide that is soluble in the medium and used it for the production of all Cu-PS complexes and as a control (See references 6,10,70).
- Fig. 1 green trace nearly undistinguishable to the blue. Differentiate clearly.
Author Response: The color in Fig. 1 of the control (polysaccharide alone) was changed. Instead of the green color a black color was used.
- Show SEM graphs of membranes and complexes.
Author Response: We have added in Fig. S1 the SEM micrograph and EDS analysis of the PS alone - Fig. S1 sections E & J. (We assume that the reviewer meant the polysaccharide since no membranes were involved).
- Fig. 2 Clearly differentiate bottom line. It is an unclear mixture.
Author Response: The bottom lines in Fig. 2 represent Cu release from monovalent Cu-PS complexes (Cu2O-PS, CuCl-PS) which are overlapping each other. We have thus changed the symbols: O and â–². Also, the figure now is without the connecting lines. We do hope it is more clear now.
- Fig. 4- define the controls.
Author Response: We have modified Fig. 4 as follows: the term “controls” was added above the 6 controls that are smaller now.
- Fig. 6A- too small to recognize bars and details.
Author Response: The images in Fig. 6A were enlarged so that the bars and details are better seen.
- Conclusions is far too long.
Author Response: The conclusions were shortened (see track changes).
- Supporting information: Draw a sketch of SEM micrographs in Figure S1.
Author Response: We have added to the Supporting information (Fig. S2) a schematic illustration of the spikes. This was also mentioned in section 2.3.
- It is hard to believe to see 5000 spikes/µM. Do you mean mm?
Author Response: The density of the spikes was measured using Gwyddion and ImageJ software and calculated per µm2 (see materials & methods, page 14, section 3.11).

Reviewer 2 Report
The work is devoted to nanotopography on the surface of a polysaccharide in a complex with mono- and divalent copper compounds. The collection strain of the red alga Porphyridium sp. served as the source of the polysaccharide. The task of the authors of the article was to test the hypothesis that the complex of copper and sulfated polysaccharide exhibits antimicrobial activity due to the action of the spikes of the complex on the cell surface. Complexes of monovalent copper (Cu2O, CuCl) and bivalent copper (CuO, CuCl2) have been obtained and studied. Gram-positive and Gram-negative bacteria as well as yeasts were used as test microorganisms.
Verification of the assumption that the antimicrobial effect is associated with mechanical damage to the cell membrane by spikes and the expiration of cell contents was carried out on a reporter strain by assessing bioluminescence.
It has been shown that complexes with monovalent copper inhibit the growth of microorganisms and the formation of biofilms better than complexes with bivalent copper. The authors attribute this to longer spikes and their high density on the surface. Verification of the assumption that the antimicrobial effect is associated with mechanical damage to the cell membrane by spikes and the expiration of cell contents was carried out on a reporter strain by assessing bioluminescence.
The enhancement of bioluminescence was interpreted as the release of cellular contents through the perforated membrane (in the reporter strain E.c. TV1061).
Many modern methods were used in the work, the experiment was performed correctly, the conclusions were substantiated. The work is of great interest to those working in this field and has general biological interest to specialists of other profiles.
The manuscript may be published without modification.
Author Response
Reviewer #2
The work is devoted to nanotopography on the surface of a polysaccharide in a complex with mono- and divalent copper compounds. The collection strain of the red alga Porphyridium sp. served as the source of the polysaccharide. The task of the authors of the article was to test the hypothesis that the complex of copper and sulfated polysaccharide exhibits antimicrobial activity due to the action of the spikes of the complex on the cell surface. Complexes of monovalent copper (Cu2O, CuCl) and bivalent copper (CuO, CuCl2) have been obtained and studied. Gram-positive and Gram-negative bacteria as well as yeasts were used as test microorganisms.
Verification of the assumption that the antimicrobial effect is associated with mechanical damage to the cell membrane by spikes and the expiration of cell contents was carried out on a reporter strain by assessing bioluminescence.
It has been shown that complexes with monovalent copper inhibit the growth of microorganisms and the formation of biofilms better than complexes with bivalent copper. The authors attribute this to longer spikes and their high density on the surface. Verification of the assumption that the antimicrobial effect is associated with mechanical damage to the cell membrane by spikes and the expiration of cell contents was carried out on a reporter strain by assessing bioluminescence.
The enhancement of bioluminescence was interpreted as the release of cellular contents through the perforated membrane (in the reporter strain E.coli TV1061).
Many modern methods were used in the work, the experiment was performed correctly, the conclusions were substantiated. The work is of great interest to those working in this field and has general biological interest to specialists of other profiles.
The manuscript may be published without modification.
Author Response: We thank the Reviewer for thorough reading of the manuscript.

Reviewer 3 Report
The manucript describes antimicrobial activity of a complex of Cu2O with the sulfated polysaccharide (PS) of the marine red microalga Porphyridium. This study continues several other works published previously. Manuscript is well written and looks fine, but there is a question - Cu2O is not soluble in water. Its solubility is so low that can not really be measured, its estimate is few ppb (https://link.springer.com/article/10.1007/s10953-011-9699-x). Here it was dissolved to 500 ppm, which is 500 000 times more. Surely presence of the PS can not influence it so much.
Also, spikes of the PS secondary structure may look sharp, but in hydrated form they must be soft and can not make physical damage to any cell. There must be some explanation to observed suppression of P. aeruginosa spread, but what is presented does not look satisfactory.
Author Response
Reviewer #3
- The manuscript describes antimicrobial activity of a complex of Cu2O with the sulfated polysaccharide (PS) of the marine red microalga Porphyridium. This study continues several other works published previously. Manuscript is well written and looks fine, but there is a question - Cu2O is not soluble in water. Its solubility is so low that cannot really be measured, its estimate is few ppb (https://link.springer.com/article/10.1007/s10953-011-9699-x). Here it was dissolved to 500 ppm, which is 500,000 times more. Surely presence of the PS cannot influence it so much.
Author Response: Indeed, Cu2O is insoluble in water, but it is soluble in acidic environment similar to that of the sulfated polysaccharide (ca pH 5.0). The pH value of the polysaccharide is presented Table 1.
The Cu-PS complexes are composed mainly of the polysaccharide and its characteristics e.g. its pH value, structure (branched), its negative charge (electrostatic interactions) and its salt composition (for example, sulfate). Also, the leakage experiments indicate that the Cu was trapped in the polysaccharide (Fig. 2).
- Also, spikes of the PS secondary structure may look sharp, but in hydrated form they must be soft and cannot make physical damage to any cell. There must be some explanation to observed suppression of aeruginosa spread, but what is presented does not look satisfactory.
Author Response: Physical disruption of cellular membranes is cited as the main reason for the anti-bacterial effects of surfaces. By modifying surface properties of the polysaccharide by spikes formation, we are interfering with cell-surface interactions. An examination of cytotoxicity provided additional support that Cu-PS has a significant cytotoxic effect on the E. coli TV1061 reporter strain responding to membrane-damaging agents (Fig. 8). The damage to the microbial cells by spikes (causing their spillage) can be seen in the attached SEM micrographs (new) that were added to the Supporting information (Fig. S5). The damage to the cells can also be seen in the SEM micrographs of Fig. 7 (without gold coating- upper left).
In addition, we present data that shows clear correlation between aspect ratio (height divided by width) of the spikes and cell death. Also, we present data showing direct correlation between the aspect ratio of the spikes and the amount of content of cells spilled out (luminescence; Fig. 9 B & C). We thus suggest that the sharp (high aspect ratio) spikes poke the cell membranes and caused the spilling of the content.
The remark of the reviewer concerning the “inability” of the spikes to cause damage are not supported by our combined and clear results:
- Damaged microbial cells after treatment with Cu2O-PS complex.
- Spilling out from the cells (luminescence).
- SEM micrographs showing damaged cells.
- Correlation between aspect ratio and bacterial cell death and luminescence.

Round 2
Reviewer 1 Report
accept
Reviewer 3 Report
Corrections and author responces look satisfactory